# Ingredients for microbial life preserved in 3.5 billion-year-old fluid inclusions

Helge Mißbach [1,6 ✉], Jan-Peter Duda [1,2,7], Alfons M. van den Kerkhof[3], Volker Lüders[4], Andreas Pack[5], Joachim Reitner[1,2] & Volker Thiel[1]

It is widely hypothesised that primeval life utilised small organic molecules as sources of carbon and energy. However, the presence of such primordial ingredients in early Earth habitats has not yet been demonstrated. Here we report the existence of indigenous organic molecules and gases in primary fluid inclusions in c. 3.5-billion-year-old barites (Dresser Formation, Pilbara Craton, Western Australia). The compounds identified (e.g., $H_2S$, COS, $CS_2$, $CH_4$, acetic acid, organic (poly-)sulfanes, thiols) may have formed important substrates for purported ancestral sulfur and methanogenic metabolisms. They also include stable building blocks of methyl thioacetate (methanethiol, acetic acid) – a putative key agent in primordial energy metabolism and thus the emergence of life. Delivered by hydrothermal fluids, some of these compounds may have fuelled microbial communities associated with the barite deposits. Our findings demonstrate that early Archaean hydrothermal fluids contained essential primordial ingredients that provided fertile substrates for earliest life on our planet.

[1] Geobiology, University of Göttingen, Göttingen, Germany. [2] "Origin of Life" Group, Göttingen Academy of Sciences and Humanities, Göttingen, Germany. [3] Applied Geology, University of Göttingen, Göttingen, Germany. [4] GFZ German Research Centre for Geosciences, Telegrafenberg, Potsdam, Germany. [5] Isotope Geology Divison, University of Göttingen, Göttingen, Germany. [6]Present address: Geobiology, University of Cologne, Cologne, Germany. [7]Present address: Sedimentology & Organic Geochemistry, University of Tübingen, Tübingen, Germany. ✉email: helge.missbach@uni-koeln.de

Primeval microbes likely required small organic molecules to act as building blocks for biomass and as catabolic substrates for heterotrophic metabolism. A potential source of such compounds includes recycled and redistributed organic matter from pre-existing biomass[1]. In addition, ample exogenous organic matter probably had been delivered to the early Earth by interplanetary dust particles and meteorites[2,3]. Experiments have also shown that organic molecules relevant for primordial life can be formed by synthesising organic compounds from inorganic atmospheric gases[4,5]. As important, endogenous synthesis and processing of organic molecules could have occurred in marine and terrestrial (i.e. hot spring) hydrothermal environments[6–12]. In such settings, organic molecules may form, or react, at elevated temperatures and pressures within the steady flow of inorganic hydrothermal chemistry (e.g. $H_2S$, $CO_2$, $H_2$[7,13]). One hypothesis on organic synthesis at hydrothermal sites suggests that the reaction of iron(II) sulfide (FeS) to pyrite ($FeS_2$) with $H_2S$ drives the reduction of $CO_2$ to organic molecules[14]. Moreover, a primordial carbon fixation mechanism involving the reaction of CO with methanethiol ($CH_3SH$) on catalytic metal (Ni-/Fe-) sulfide surfaces could be demonstrated in the laboratory under hydrothermal conditions. This experiment produced an activated form of acetic acid that represents a plausible building block for further organic synthesis, for example, into acyl lipids[15]. As yet, however, such distinctive organic molecules have not been found in rocks that directly testify to the emergence of life on our planet.

The c. 3.5 billion-year-old Dresser Formation (Pilbara Craton, Western Australia) is one of the most important windows into hydrothermal habitats on early Earth[16,17]. The rocks are only mildly metamorphosed (prehnite-pumpellyite to lower greenschist facies[18,19]) and still preserve numerous putative biosignatures, including stromatolites[10,17,20,21], microfossils[22,23], and isotopic anomalies[23–27]. Further, cherts ($SiO_2$) and barites ($BaSO_4$) of the Dresser Formation contain kerogenous organic material of supposedly biological origin[1,22,26,28]. Detailed field mapping, petrographic observations, and mineralogical analyses revealed that the Dresser Formation was formed in a hydrothermal setting, most likely a volcanic caldera[10,16,17]. Thus, it appears plausible that organisms in the Dresser environments grew chemotrophically, fuelled by hydrothermal fluids that delivered inorganic and organic substrates. Indeed, stable carbon and sulfur isotopic anomalies indicate methanogenic and sulfur-disproportionating microbes as key players in these early microbial communities[16,27,29], although the exact metabolisms still await further evidence and testing.

Cherts and barites of the Dresser Formation contain abundant primary fluid inclusions[29–33], that is, fluids and/or gases entrapped in minerals[34]. These fluid inclusions represent a valuable archive, as their chemistry can potentially be preserved for billions of years[29–32,35–37]. Barite appears to be a particularly robust host mineral because of its low solubility and high stability under a wide range of pressure, temperature and redox conditions[38,39]. Therefore, fluid inclusions in the Dresser barites are excellent candidates in the search for organic molecules that once supported microbial life. Previous work identified $H_2O$, $CO_2$, $H_2S$, and minor $CH_4$ as the main inorganic constituents of the fluid inclusions in Dresser barites[32,33]. However, the content of organic molecules, potential key ingredients for early life, is as yet unknown.

Here we report on the presence of biologically-relevant primordial organic molecules in primary fluid inclusions trapped in barites of the c. 3.5 billion-year-old Dresser Formation. To explore the full range of volatiles, we combined gas chromatography–mass spectrometry (GC–MS), microthermometry, fluid inclusion petrography, and stable isotope analysis. Our findings reveal an intriguing diversity of organic molecules with known or inferred metabolic relevance and provide a strong clue as to how ancient hydrothermal fluids sustained microbial life ~3.5 billion years ago.

## Results

**Field and petrographic observations.** The Dresser Formation contains thick barite units with colours ranging from white and grey to black. Black barites exhibit coarse crystalline textures and yield a strong $H_2S$ odour when freshly crushed. The sedimentary black barite studied here was sampled in the Dresser mine, where it was interbedded with originally sulfidic[17,40] stromatolites (Fig. 1). Field and petrographic evidence clearly suggest a primary origin of the barite (e.g. no progressive replacement of stromatolite interbeds, no relicts of potential precursor materials within the barite: Figs. 1, 2). Thin section analysis revealed the presence of abundant primary and rare secondary inclusions (Fig. 2a–c). Most primary fluid inclusions are small (c. 10 μm), translucent, and often oriented parallel to planes of barite crystals, thereby tracing succeeding growth phases. Morphologies of some fluid inclusions indicated necking down, which is a typical modification under stress conditions after crystallisation[34]. These inclusions are typically stretched and may split up in segments that then usually show different composition and density.

**Fluid inclusion classification and composition.** The fluid inclusions were analysed optically on a heating-freezing stage and by Raman spectroscopy. The black barites contain aqueous carbonic-sulfuric and non-aqueous carbonic-sulfuric fluid inclusions (hereafter, aqueous, and non-aqueous, respectively; Table 1). Aqueous inclusions show highly variable water volume fractions of 0.1–1. At room temperature, they typically exhibit a double meniscus, indicating the presence of three phases: water + another ($CO_2$–$H_2S$-rich) liquid + vapour (Fig. 2d). In some cases, the other liquid is only visible during cooling runs. In comparison, non-aqueous fluid inclusions usually contain a $CO_2$–$H_2S$-rich liquid and a vapour phase (Fig. 2e; Table 1), although the liquid phase is sometimes absent at room temperature.

Both types of fluid inclusions typically contain solid daughter phases (Fig. 2d, e; Table 1). Aqueous inclusions usually contain strontianite ($SrCO_3$) and sulfur as daughter crystals. Varieties with pure $CO_2$ in the vapour phase (volume fractions of c. 0.9) may additionally include anatase ($TiO_2$), pyrite, and possibly also halite (NaCl). In non-aqueous inclusions, typical daughter phases are sulfur, kerogen and, in few cases, halite.

The main gas components in both fluid inclusion types are $CO_2$ and $H_2S$, accompanied by minor amounts of $CH_4$, $N_2$, and COS (carbonyl sulfide) (Fig. 3). Aqueous fluid inclusions contain less $H_2S$ than non-aqueous fluid inclusions (0–24 mol% and 21–36 mol%, respectively). Furthermore, aqueous fluid inclusions typically enclose up to 1 mol% $N_2$, which is not present in non-aqueous fluid inclusions. Instead, non-aqueous fluid inclusions additionally contain small amounts of $CH_4$ (<2 mol%) (Fig. 3; Table 1).

**Fluid inclusion microthermometry.** Aqueous fluid inclusions typically reveal liquid compositions ranging from pure water to more saline solutions with 14 wt% NaCl-equivalents. Higher salinities of up to 25 wt% NaCl-equivalents are rare. The corresponding ice melting temperatures vary between 0 °C and −26 °C (peak at −7 °C; Supplementary Fig. 1). Aqueous fluid inclusions form clathrates upon freezing and subsequent melting between 7 °C (pure $CO_2$) and 20 °C ($H_2S$-rich). Total homogenisation temperatures [Th (total)], describing the minimum temperature of fluid entrapment, range from 100 to 195 °C, with a maximum

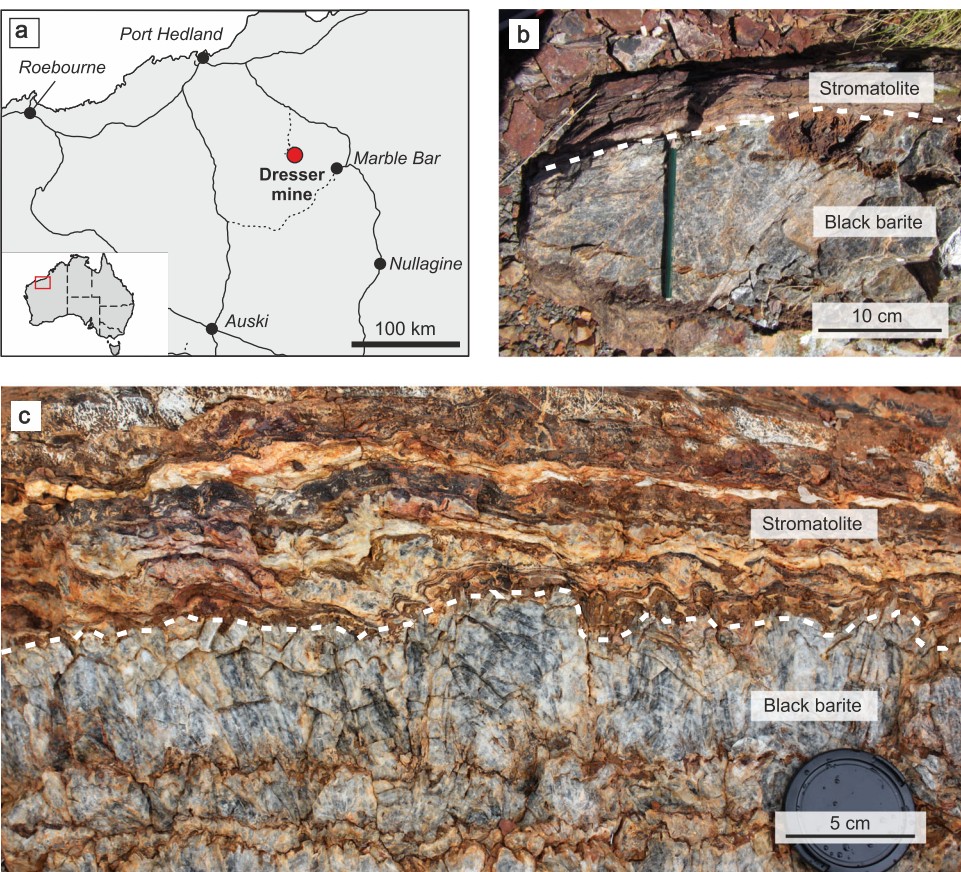

**Fig. 1 Study area and field evidence.** Location of the Dresser mine in Western Australia near Marble Bar (**a**) and black barite associated with originally sulfidic stromatolites at the sampling site (**b**) and in the working area (**c**). The close association between the inclusion-bearing black barites and stromatolites suggests that hydrothermal fluids might have influenced ancient microbial communities.

between 110 and 150 °C (Supplementary Fig. 1). Most fluid inclusions decrepitate at temperatures >230 °C.

Non-aqueous fluid inclusions show Th ($CO_2$-$H_2S$) between 16 and 38 °C (Supplementary Fig. 1). Those containing higher concentrations of $H_2S$ typically homogenise at the higher end of this range, that is, above the critical temperature of $CO_2$ (31.1 °C). Phases usually homogenise to liquid, and only rarely to the gas or critical phase. During cooling runs, the subsequent melting of solid $CO_2$ and $H_2S$ can be observed at lower temperatures compared to the pure compounds (−56.6 °C and −83.6 °C, respectively) (Supplementary Fig. 1).

Our data demonstrate that the majority of aqueous and non-aqueous inclusions formed during crystal growth (i.e. primary inclusions). Thus, fluids must have been immiscible at the time of encapsulation, and experienced identical trapping and homogenisation temperatures (i.e. heterogeneous trapping). Therefore, no pressure correction is necessary.

**Thermal decrepitation/desorption (TD) GC–MS.** Online analyses of black barite fragments using TD-GC–MS yielded high amounts of $CO_2$, $H_2S$, and $H_2O$, thus confirming results from Raman analysis on fluid inclusions (Fig. 4, Table 1). Furthermore, COS, $CS_2$, $SO_2$, together with various organic molecules containing oxygen (aldehydes, ketones, acetic acid, oxolane) and sulfur (thiophene, thiols) were detected, accompanied by minor amounts of benzene (Fig. 4, Supplementary Fig. 2, Table 2). The diversity and intensity of compounds was considerably higher in the 250 °C than in the 150 °C experiment (Table 2). This finding is consistent with the microthermometry data revealing that most fluid inclusions remain intact up to ~230 °C.

**Solid phase micro extraction (SPME) GC–MS.** Offline analysis using SPME-GC–MS revealed numerous organic molecules containing oxygen (aldehydes, ketones, acetic acid, oxolane) and/or sulfur (thiophene, thiols, organic polysulfanes), along with some aromatic hydrocarbons (e.g. benzene, alkylbenzenes; Fig. 5, Supplementary Fig. 2, Table 2). Compounds detected with both analytical techniques showed a lower abundance in SPME-GC–MS as compared to TD-GC–MS at 250 °C. On the other hand, SPME-GC–MS yielded a considerably greater diversity of compounds, especially in the higher molecular weight range (Figs. 4, 5). The absence of $CO_2$ and $H_2S$ in the SPME-GC–MS runs is due to an analytical bias, as these compounds do not adsorb onto the SPME fibre.

**Organic carbon content and stable isotopes (C, O).** The mean total organic carbon (TOC) content of the black barite is 0.31 wt % ($N = 5$; standard deviation 0.002). Stable carbon isotope analysis revealed a mean $\delta^{13}C_{TOC}$ value of −27.6 ± 0.6 ‰. Offline analysis revealed $\delta^{13}C_{CO2}$ and $\delta^{18}O_{CO2}$ values of −10.0 ± 0.3 ‰ and 34.1 ± 0.6 ‰, respectively. Online analyses yielded $\delta^{13}C_{CO2}$ values ranging from −14.3 to −8.9 ± 0.3 ‰ for black barites (mean = −10.3 ‰) and from −8.6 to −4.0 ± 0.3 ‰ for grey barites (mean = −6.3 ‰). Thus, black barites are consistently more depleted in $^{13}C$ than their grey counterparts (Fig. 6). In all cases, $CH_4$ and $N_2$ contents were too low for stable isotope analyses (<2 mol %).

## Discussion

**Data integrity.** Black barites studied here classify as primary hydrothermal sediments that precipitated from discharging fluids.

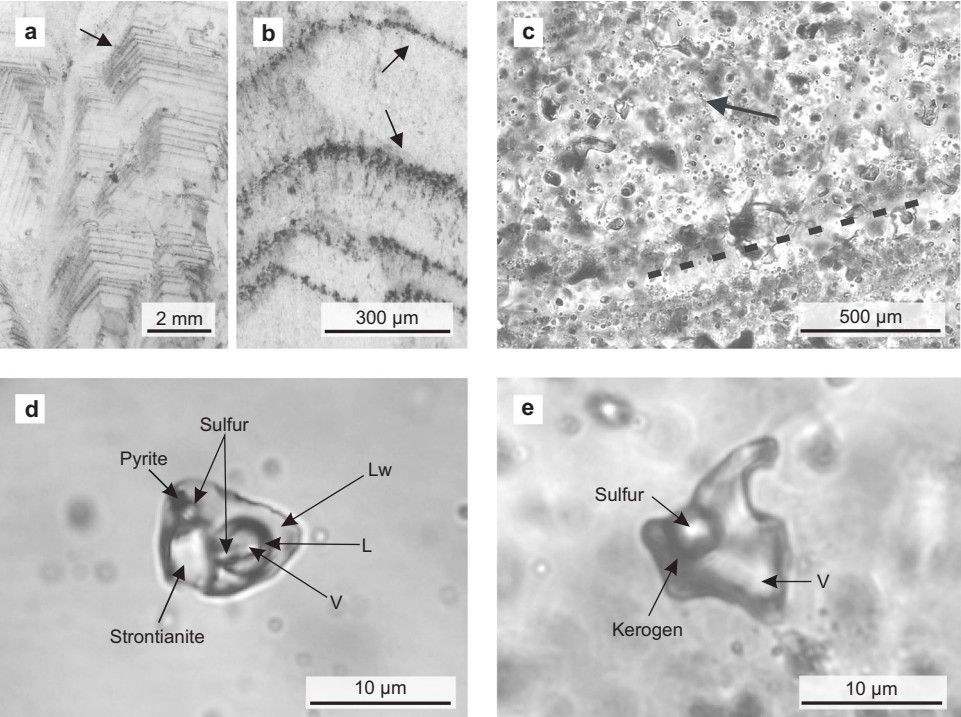

**Fig. 2 Fluid inclusions in representative black barites from the Dresser mine. a, b** Thin section images (reflected light) showing primary fluid inclusion trails parallel to barite crystal growth bands (marked by black arrows). **c** Thin section image (transmitted light) showing primary fluid inclusions which are dispersed or oriented parallel to barite crystal growth bands (exemplified by dashed line). The image also shows a minor secondary inclusion trail (marked by black arrow). **d** Thick section image (transmitted light) of an aqueous carbonic-sulfuric fluid inclusion containing three volatile phases (including $H_2S$), plus pyrite, native sulfur, and strontianite as solid phases. **e** Thick section image (transmitted light) of a non-aqueous fluid inclusion bearing a vapour phase, native sulfur, and kerogen. These fluid inclusions are usually rich in $H_2S$. V vapour/gas, Lw liquid $H_2O$, L other liquid (e.g. $CO_2$). Organic compounds and gases preserved in these primary fluid inclusions could have provided a substrate to primordial microbial life in the Dresser Formation.

This interpretation is additionally supported by the facts that (i) the originally sulfidic[17,40] stromatolite interbeds are still largely intact and show no indications for a progressive replacement by barite (Fig. 1b, c) and (ii) that the barite does not contain relicts of potential precursor materials (Fig. 2). Our observations are therefore consistent with earlier studies that argued for a primary, synsedimentary origin of the Dresser barites analysed herein (i.e. precipitation in surface environments linked to hydrothermal activity)[10,16,32,41,42].

Barite is highly chemically stable under a wide range of geological conditions[38,39]. Hence, barite-hosted fluid inclusions can preserve information on the original composition of hydrothermal fluids. The black and grey barites from the Dresser Formation primarily grew as coarse crystals and contain abundant primary fluid inclusions (see also refs. [17,32,39]). Most fluid inclusions show no indication of post-entrapment modification. The results are reproducible and Th values [100–195 °C for Th (total)] are internally consistent for different coevolutionary fluid inclusions. The measured Th is in line with (i) formation temperatures estimated for coexisting cherts (100–200 °C)[26], (ii) other reported Th from fluid inclusions in Dresser barites and cherts (110–365 °C)[32], and (iii) maximum formation temperatures of barite-hosted fluid inclusions in a modern hydrothermal system (JADE hydrothermal field, 150–200 °C)[43].

The aqueous and non-aqueous fluid inclusions distinguished herein appear to include those described in earlier studies[32,33]. Particularly key-characteristics such as sizes (5–30 µm), ice and clathrate melting temperatures (−7.5 to −0.6 °C and −0.9 to 19.2 °C, respectively), and the fundamental volatile inventories ($CO_2$, $H_2O$, $H_2S$, $CH_4$) are all remarkably similar. A notable exception is the presence of trace amounts of $N_2$ in some of the aqueous fluid inclusions (Table 1, Fig. 3), which has not been reported previously.

The presence of aqueous and non-aqueous fluid inclusions (Fig. 3; Table 1) can be explained by the presence of two coexisting fluids at the time of trapping as a result of phase separation from boiling fluids during cooling (effervescence)[34]. Hence, the major fluid composition of the black barites can be considered primary. However, there are indications that a few fluid inclusions were locally modified immediately after emplacement (e.g. necking down after crystallization[34]), explaining the wide variations observed in Th (total). This information is not relevant to the interpretation of the fluids as being primary, because they would be trapped again instantly with their overall composition remaining unchanged.

Organic molecules detected by GC–MS are derived from the fluid inclusions as evidenced by (i) clean pre-analysis blanks, (ii) retrieval of products exclusively after grinding of barite, (iii) reproducibility of the results from five TD-GC–MS and seven SPME-GC–MS experiments, (iv) presence of highly volatile compounds in GC–MS analyses, (v) consistency of data obtained by independent analytical techniques (Raman spectroscopy vs. GC–MS), (vi) temperature dependency of product yields, meaning that higher temperature analyses above the decrepitation temperature of fluid inclusions result in higher abundances (TD-GC–MS 150 °C vs. TD-GC–MS 250 °C), and (vii) absence of molecular contamination indications. Together, these multiple lines of evidence strongly suggest that the analysed compounds derived from the fluid inclusions, while a minor contribution of organic compounds from the rock matrix cannot entirely be ruled out. This result adds to earlier studies, which demonstrated that fluid inclusions form closed systems that can preserve molecules even in billion-year-old metamorphic rocks[36,37,44,45].

**Table 1 Overview of gas composition and solid phases in barite-hosted fluid inclusions as determined by Raman analysis.**

| Fluid inclusion | Phases at $T_{room}$ | Water vol. frac. | Gas composition (mol%) | | | | | Raman-active solids | | | | |
|---|---|---|---|---|---|---|---|---|---|---|---|---|
| | | | $CO_2$ | $H_2S$ | $CH_4$ | $N_2$ | COS (carbonyl sulfide) | Kerogen | Anatase | Pyrite | Strontianite | Sulfur |
| Aqueous carbonic-sulfuric inclusions (± strontianite ±sulfur) | | | | | | | | | | | | |
| 2A_01 | Lw + V + S (not Raman-active) | 0.99 | n.d. | | | | | | | | | |
| 3F-4_04 | Lw + V | 0.99 | 100 | | | | | | | | | |
| 3A_05 | L + V + S1 + S2 + opaque | 0.9 | 100 | | | | | | X | X | | |
| 3F-4_03 | Lw + L + V + S | 0.6 | 99.2 | 0.8 | | | | | | | X | |
| 3F-4_01 | Lw + V | 0.8 | 96.2 | 3.1 | | | | | | | | |
| 1D-1_01 | Lw + V + S | 0.9 | 93.8 | 6.2 | | 0.7 | | | | | | X |
| 3F-4_02 | Lw + V + S | 0.9 | 93.4 | 6.6 | | | | | | | X | X |
| 3_01 | Lw + V + S1 + S2 + S3 + S4 | 0.6 | 91.6 | 8.4 | | | | | | | X | X |
| 3D_01 | Lw + L + V + S | 0.3 | 87.0 | 13.0 | | | X | | | | X | |
| 3D_04 | (Lw) + L + V + S1 + S2 | 0.1 | 86.0 | 14.0 | | | | | | | X | X |
| 3F-4_05 | Lw + V + S1 + S2 + S3 | 0.95 | 85.6 | 14.4 | | (X) | | | | X | X | X |
| 1D-1_02 | Lw + V + S | 0.9 | 85.3 | 13.6 | | (X) | | | | | | X |
| 3D_03 | (Lw) + L + V + S1 + S2 | 0.1 | 84.3 | 15.2 | | 0.4 | ? | | | | X | X |
| 3-D_05 | Lw + L + V + S1 + S2 + S3 + opaque | 0.7 | 84.0 | 16.0 | | | X | | | ? | X | ? |
| 3D_02 | (Lw) + L + V | 0.1 | 81.0 | 19.0 | | | ? | | | | | |
| 1A-2_08 | Lw + L2 + V + S1 | 0.9 | 76.0 | 24.0 | | | ? | | | | X | |
| Non-aqueous carbonic-sulfuric inclusions (± sulfur ±kerogen) | | | | | | | | | | | | |
| 1A-2_04 | L + V + S1 + S2 | « | 78.0 | 21.6 | 0.4 | | | X | | | | X |
| 1A-2_06 | L + V + S | « | 78.0 | 21.6 | 0.4 | | (X) | X | | | | |
| 1A-2_05 | L + V + S1 + S2 | « | 77.5 | 23.0 | 0.5 | | | (X) | | | | X |
| 1A-2_07 | L + S1 | « | 76.0 | 23.5 | 0.5 | | ? | X | | | | X |
| 1C-1_01 | L + V + S | « | 64.9 | 33.7 | 1.4 | | | | | | | X |
| 1C-1_02 | L + V + S | « | 62.7 | 36.4 | 0.9 | | | | | | | X |

*Lw* liquid water, *L* other liquid (e.g. $CO_2$), *V* vapour, *S* solid, *vol. frac.* volume fraction.

Organic molecules detected by TD-GC–MS and SPME-GC–MS display major differences in diversity and abundance (compare Figs. 4 and 5). SPME probably provides a more authentic picture of the compounds contained in the fluid inclusions, because no heating to >50 °C is applied before GC–MS analysis. In contrast, TD resulted in abundant $SO_2$ formation during heating to higher temperatures (Table 2, 250 °C experiment), reflecting thermally driven artefact formation by reaction of the components in the interior of the fluid inclusions. Additionally, and even more important, the mild SPME offline approach can be applied on much greater sample amounts (g vs.

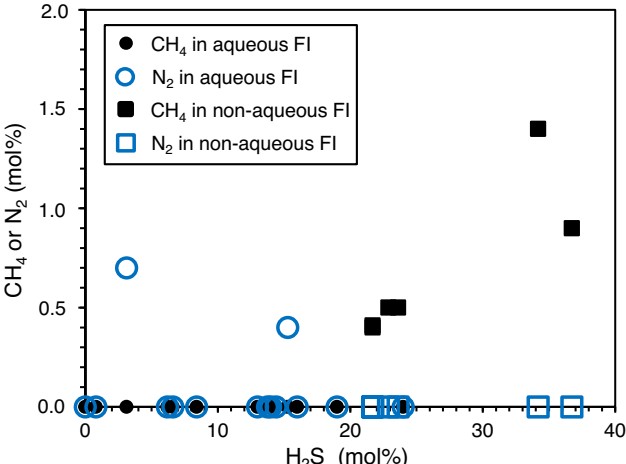

**Fig. 3 Gas compositions of fluid inclusions in black barites as measured by Raman analysis.** 0 % $H_2S$ implies that the fluid inclusion largely contains $CO_2$. FI fluid inclusions.

mg), resulting in detectable yields of trace compounds that are indiscernible with the TD approach.

**Nature and origin of fluids.** The Dresser Formation formed in a hydrothermal environment[16,17]. Hence, compounds entrapped in barite-hosted fluid inclusions may have been derived from abiotic sources. Indeed, gaseous compounds such as $SO_2$, $CO_2$, $H_2S$, COS, $CS_2$, and (methylsulfanyl)methane are known to be delivered to surface environments via volcanic outgassing[46,47]. Functionalized lipid-like organic molecules such as ketones, aldehydes, carboxylic acids, and alcohols can be formed by Fischer–Tropsch-type processes under hydrothermal conditions[6–9]. Further compounds of possibly abiotic origin are acetic acid and organic sulfur molecules (e.g. thiols, organic polysulfanes). These molecules may be synthesized in the presence of sulfide catalysts[15] and with $CS_2$ or $CO_2$ as a carbon source[48,49]. Extraterrestrial delivery by meteorites could have provided an additional source for many of the observed compounds (e.g. COS, $CS_2$, $H_2S$, methanethiol, benzaldehyde, acetic acid, benzene, toluene, various aldehydes, and ketones)[2,3,50–52].

While many compounds observed in the barite-hosted fluid inclusions from the Dresser Formation are consistent with an abiotic origin, the Dresser Formation also contains a variety of evidence for life[1,10,17,20–25,27,29]. Thus, biology is another potential source for the observed compounds. In fact, organisms synthesise most lipids on modern Earth, and $\delta^{13}C$ signatures of kerogen in the black barite (c. –28 ‰) are in good accordance with biological carbon fixation[53]. Furthermore, compounds such as $H_2S$, COS, $CS_2$, (methylsulfanyl)methane, (methyldisulfanyl)methane, and thiols are typically formed during microbial sulfur cycling in modern environments[54–57], and there is isotopic evidence for the presence of sulfur-processing metabolisms during Dresser times[24,25,27].

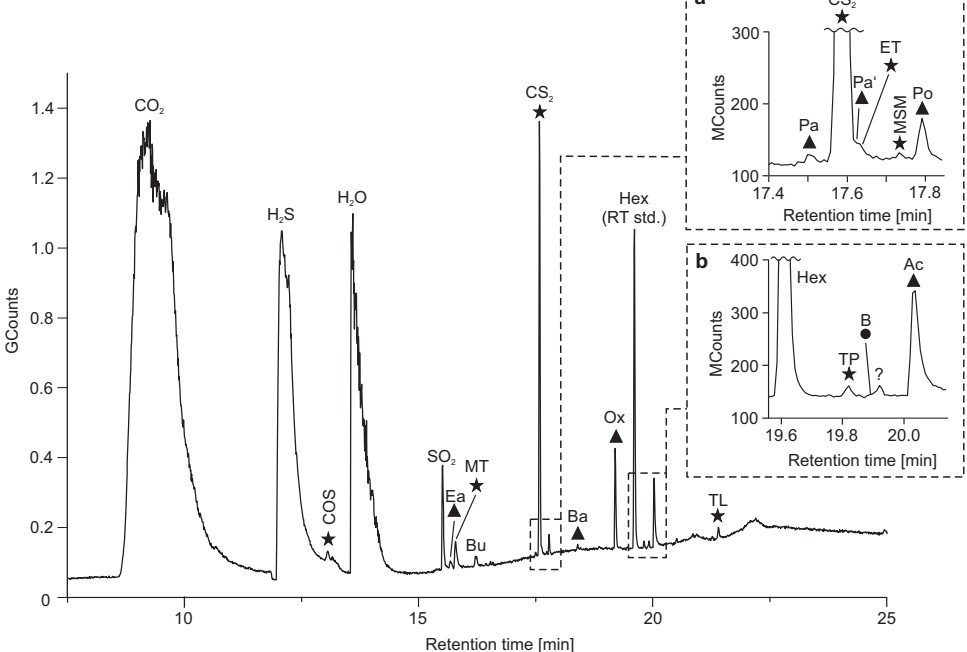

**Fig. 4 Total ion current chromatogram of volatile compounds from black barite fluid inclusions as detected by thermal decrepitation/desorption GC–MS (TD-GC–MS) analysis at 250 °C.** Inserts **a**, **b** represent enlargements of respective areas in the chromatogram marked by dashed lines. Triangles denote oxygen-bearing compounds, circles denote aromatic hydrocarbons and stars denote sulfur-bearing compounds. n-Hexane (Hex) was used as a retention time standard (RT std.). COS carbonyl sulfide, Ea ethanal, MT methanethiol, Bu but-1-ene, Pa prop-2-enal, Pa' propanal, ET ethanethiol, MSM (methylsulfanyl)methane, Po propan-2-one, Ba but-2-enal, Ox oxolane, TP thiophene, B benzene, Ac acetic acid, TL thiolane. Note the presence of methanethiol and acetic acid, the stable building blocks of activated acetic acid.

**Table 2 Overview of volatile organic compounds from TD-GC–MS and SPME-GC–MS analysis.**

| Peak label | Compound | Level of confidence[a] | Chemical formula | Key ions (m/z) | RT[b] | Rel. abundance TD [%][c] 150 °C | 250 °C | Rel. abundance SPME [%][c] Median | SD |
|---|---|---|---|---|---|---|---|---|---|
| $CO_2$ | Carbon dioxide | 2 | $CO_2$ | 44 | 9.27* | High | High | | |
| $H_2S$ | Hydrogen sulfide | 2 | $H_2S$ | 34 | 12.09* | High | High | | |
| COS | Carbonyl sulfide | 2 | COS | 60 | 11.51 | 24.48 | 4.56 | 4.59 | 1.41 |
| $H_2O$ | Water | 2 | $H_2O$ | 18 | 12.93* | High | High | high | 0,00 |
| $SO_2$ | Sulfur dioxide | 2 | $SO_2$ | 48, 64 | 15.51 | | 14.31 | | |
| Ea | Ethanal | 2 | $C_2H_4O$ | 44 | 15.58 | | 1.40 | 1.91 | 1.75 |
| MT | Methanethiol | 2 | $CH_4S$ | 48 | 15.70 | 7.80 | 5.04 | 3.97 | 0.46 |
| Bu | But-1-ene | 1 | $C_4H_8$ | 56 | 16.21 | tr | 1.84 | tr | |
| Eo | Ethanol | 3 | $C_2H_6O$ | 45 | 17.26 | | tr | tr | |
| Pa | Prop-2-enal | 2 | $C_3H_4O$ | 56 | 17.55 | | 0.35 | 0.33 | 0.00 |
| $CS_2$ | Methanedithione | 2 | $CS_2$ | 76 | 17.63 | 43.20 | 45.80 | 9.08 | 1.63 |
| Pa' | Propanal | 3 | $C_3H_6O$ | 58 | 17.68 | | 0.35 | 0.54 | 0.42 |
| ET | Ethanethiol | 2 | $C_2H_6S$ | 29, 47, 62 | 17.70 | tr | 0.43 | 0.69 | 0.27 |
| MSM | (Methylsulfanyl)methane | 2 | $C_2H_6S$ | 47, 62 | 17.79 | tr | 0.13 | 1.56 | 0.49 |
| Po | Propan-2-one | 2 | $C_3H_6O$ | 43, 58 | 17.84 | 3.89 | 1.87 | 7.28 | 2.48 |
| Ba | But-2-enal | 3 | $C_4H_6O$ | 70 | 18.46 | | 0.64 | 0.87 | 0.00 |
| Ox | Oxolane[d] | 2 | $C_4H_8O$ | 72 | 19.27 | tr | 11.28 | 4.71 | 3.64 |
| Bo | Butan-2-one | 3 | $C_4H_8O$ | 72 | 19.38 | | tr | 3.06 | 1.23 |
| Hex | n-Hexane (RT standard) | | $C_6H_{14}$ | | 19.68 | | | | |
| TP | Thiophene | 2 | $C_4H_4S$ | 84 | 19.89 | tr | 0.64 | 0.53 | 0.04 |
| B | Benzene | 2 | $C_6H_6$ | 78 | 19.97 | | 0.38 | 0.52 | 0.33 |
| Ac | Acetic acid | 3 | $C_2H_4O_2$ | 60 | 20.10 | 20.63 | 8.89 | 1.71 | 0.29 |
| MB | 3-methylbutan-2-one | 3 | $C_5H_{10}O$ | 86 | 20.44 | | | 0.69 | 0.00 |
| Mxp | 1-Methoxypropan-2-ol | 3 | $C_4H_{10}O_2$ | 45,47 | 20.72 | | | 4.01 | 0.00 |
| Pe | Pentanal | 3 | $C_5H_{10}O$ | 44,58 | 20.73 | | | tr | |
| MDSM | (Methyldisulfanyl)methane | 2 | $C_2H_6S_2$ | 94 | 20.91 | | | 7.70 | 3.58 |
| To | Toluene | 2 | $C_7H_8$ | 91 | 21.36 | | | 4.05 | 1.08 |
| TL | Thiolane | 2 | $C_4H_8S$ | 60, 88 | 21.41 | | 1.20 | | |
| MP | 4-Methylpentan-2-one | 3 | $C_6H_{12}O$ | 43, 58, 85 > 100 | 21.63 | | | 5.23 | 0.90 |
| Ha | Hexanal | 3 | $C_6H_{12}O$ | 56, 82 | 22.17 | | | 2.53 | 0.15 |
| MEDS | (Methyldisulfanyl)ethane | 2 | $C_3H_8S_2$ | 108 | 22.34 | | | 1.31 | 0.38 |
| Xy I | p-Xylene | 2 | $C_8H_{10}$ | 91, 106 | 22.91 | | | tr | |
| Xy II | m-Xylene | 2 | $C_8H_{10}$ | 91, 106 | 22.99 | | | 1.04 | 0.32 |
| Pac | 1-Methoxyprop-2-yl acetate | 1 | $C_6H_{12}O_3$ | 72, 87 | 23.19 | | | 1.15 | 0.06 |
| Xy III | o-Xylene | 2 | $C_8H_{10}$ | 91, 106 | 23.37 | | | tr | |
| Sty | Styrene | 1 | $C_8H_8$ | 91, 104 | 23.41 | | | tr | |
| CH | Cyclohexanone | 3 | $C_6H_{10}O$ | 98 | 23.66 | | | 1.97 | 1.28 |
| Hp | Heptanal | 3 | $C_7H_{14}O$ | 55, 70 | 24.27 | | | 1.09 | 0.07 |
| BA | Benzaldehyde | 2 | $C_7H_6O$ | 77, 106 | 25.09 | | 0.89 | 16.23 | 3.78 |
| MTSM | (Methyltrisulfanyl)methane | 2 | $C_2H_6S_3$ | 126 | 25.18 | | | 11.44 | 7.07 |
| TMB I | 1,3,5-Trimethyl benzene | 1 | $C_9H_{12}$ | 105, 120 | 25.23 | | | tr | |
| TMB II | 1,2,4-Trimethyl benzene | 2 | $C_9H_{12}$ | 105, 120 | 25.35 | | | tr | |
| MH | 6-Methylheptan-3-one | 1 | $C_8H_{16}O$ | 72, 99 | 25.95 | | | 2.03 | 0.00 |
| TMB III | 1,2,3-Trimethyl benzene | 2 | $C_9H_{12}$ | 105, 120 | 26.04 | | | 2.03 | 0.68 |

See also Supplementary Fig. 2 for molecular structures.
RT retention time, TD thermal decrepitation/desorption, SPME solid phase micro extraction, SD standard deviation, tr traces.
[a]Level of confidence: 3 = verified by standard, 2 = clear identification with reference spectra, 1 = tentative identification.
[b]RT marked by asterisks (*) are from TD-GC–MS analysis, other RT from SPME-GC–MS experiments. From 15 min RT in both methods are similar.
[c]Compounds present in traces (tr) and main components ($CO_2$, $H_2S$, $H_2O$) were not included in relative abundance calculations.
[d]Small amounts of oxolane were identified in SPME-GC–MS blanks (see Supplementary Fig. 4).

Taken together, it is likely that the barite-hosted fluid inclusions contain mixtures of various abiotic and biotic compounds. Such contributions from different sources would plausibly explain the contrasting $\delta^{13}C$ signatures of $CO_2$ in grey and black barites (Fig. 6). $CO_2$ released from grey barites exhibits a mean $\delta^{13}C$ value of –6.3 ‰, which might be indicative of a magmatic source (typically between −2 and −8 ‰[58–60]). In contrast, lower $\delta^{13}C$ values of −10.3 ‰ in $CO_2$ from black barites might fingerprint a biomass-derived carbon component that had been converted to $CO_2$ via bacterial and/or thermochemical sulfate reduction[61,62] before it was absorbed and transported by

fluids. The processing, re-distribution, and mixing of fluids from different sources[17,32] is well known from modern and ancient hydrothermal systems (hydrothermal pump[1,63]).

**Ingredients for early life in 3.5 Ga old fluid inclusions.** It is widely assumed that hydrothermal processes fuelled primeval life on Earth[13,16], but it is difficult to pinpoint the exact nature of such relationships in the Archaean rock record. The fluid inclusion-bearing black barites are interbedded with stromatolites (Fig. 1b, c; see also refs. [17,21]), suggesting that hydrothermal fluids

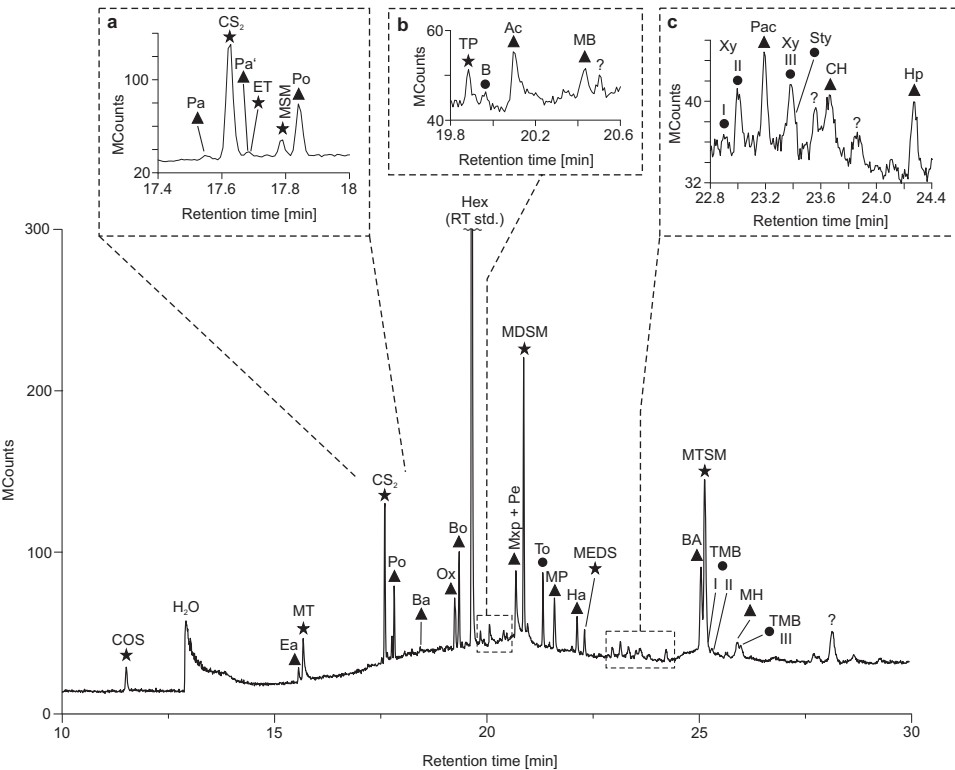

**Fig. 5 Total ion current chromatogram of volatile compounds from black barite fluid inclusions as obtained by solid phase micro extraction GC–MS (SPME-GC–MS).** Inserts **a**–**c** represent enlargements of respective areas in the chromatogram marked by dashed lines. Triangles denote oxygen-bearing compounds, circles denote aromatic hydrocarbons and stars denote sulfur-bearing compounds. *n*-Hexane (Hex) was used as a retention time standard (RT std.). COS carbonyl sulfide, Ea ethanal, MT methanethiol, Pa prop-2-enal, Pa' propanal, ET ethanethiol, MSM (methylsulfanyl)methane, Po propan-2-one, Ba but-2-enal, Ox oxolane, Bo butan-2-one, TP thiophene, B benzene, Ac acetic acid, MB 3-methylbutan-2-one, Mxp 1-methoxypropan-2-ol, Pe pentanal, MDSM (methyldisulfanyl)methane, To toluene, MP 4-methylpentan-2-one, Ha hexanal, MEDS (methyldisulfanyl)ethane, Xy I *p*-xylene, Xy II *m*-xylene, Pac 1-methoxyprop-2-yl acetate, Xy III *o*-xylene, Sty styrene, CH cyclohexanone, Hp heptanal, BA benzaldehyde, MTSM (methyltrisulfanyl)methane, TMB I 1,3,5-trimethyl benzene, TMB II 1,2,4-trimethyl benzene, MH 6-methylheptan-3-one, TMB III 1,2,3-trimethyl benzene. Note the higher diversity of compounds as compared to thermal decrepitation/desorption analysis (Fig. 4). Oxygen- and sulfur-bearing organic compounds may have provided substrates for microbial life in the Dresser Formation.

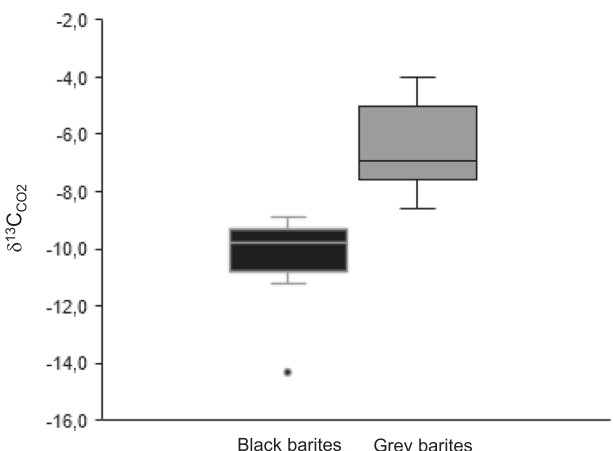

**Fig. 6 Distribution of stable carbon isotope signatures of $CO_2$ from black and grey barite fluid inclusions.** Reproducibility of the stable isotope measurements is 0.3 ‰. A total of 11 black barite samples and 11 grey barite samples was analysed. The relatively low $\delta^{13}C$ values in the black barites possibly reflect the addition of a biomass-derived carbon component to the fluids.

may have influenced the ancient microbial communities[40]. Indeed, many compounds discovered in the barite-hosted fluid inclusions (e.g. COS, $CS_2$, acetic acid, (methylsulfanyl)methane, (methyldisulfanyl)methane, thiols, $CH_4$; Figs. 4, 5) would have provided ideal substrates for the sulfur-based and methanogenic microbes previously proposed as players in the Dresser environment[24,25,27,29,40,54,55,57,64–66]. For instance, acetic acid may have fuelled acetoclastic methanogenesis, while organic sulfides such as methanethiol and (methylsulfanyl)methane might have served as substrates for fermenting methanogenic bacteria[67]. This hypothesis is in full agreement with isotopic evidence indicating the existence of methanogenic and sulfur-cycling microbes in Dresser environments[24,27,29,40]. The activity of sulfate reducing or sulfur disproportioning bacteria could also account for the presence of abundant pyrite in the originally sulfidised Dresser stromatolites[21,24,27,40,68]. Thus, our findings provide a strong clue that microbial life associated with the black barites of the Dresser Formation was (partly at least) fuelled by hydrothermal fluid flow.

In addition to potential nutrients and/or substrates, hydrothermal fluids captured in the Dresser fluid inclusions contain molecules closely related to putative key agents in the emergence of life. It has been proposed that CO and methanethiol can react in the presence of catalytic metallic sulfides to methyl

thioacetate[15]. This compound, also known as activated acetic acid, was proposed as being important for the formation of lipids under primordial conditions and as an energy source for early microbial metabolisms[13,69]. Whereas this highly energetic molecule is readily hydrolysed[70] and cannot be preserved over geological time, our data evidence the presence of its stable building blocks, methanethiol and acetic acid, in the Dresser fluids (Figs. 4, 5). In other words, essential ingredients of methyl thioacetate, a proposed critical agent in the emergence of life, were available in the Dresser environments.

Our data provide the first detailed picture of the organic composition of primordial fluids that had evidently been available for the ancient microbes c. 3.5 Ga ago. These fluids delivered ample catabolic substrates for chemoheterotrophic metabolisms. In addition, they might have conveyed the building blocks for chemoautotrophic carbon fixation and, thus, anabolic uptake of carbon into biomass. Taken together, our data strongly support the idea that hydrothermal fluids supplied a fertile substrate for early microbial life on Earth.

## Methods

**Sampling and sample preparation**. Black barite was sampled from a freshly exposed cut wall in the Dresser mine (Pilbara Craton, Western Australia; 21° 09' 05.2'' S, 119° 26' 15.3'' E; Fig. 1). Fresh grey and white barite samples were collected in the immediate surroundings of the Dresser mine and analysed for comparison. During the sampling procedure disposable nitrile gloves were worn. Individual samples were taken using a carefully pre-cleaned geological hammer and chisel. Only large pieces (>10 cm) that were devoid of any weathering features down to the hand-lens scale (e.g. cracks and fractures, biological surface coloni-zation) were selected. Immediately after recovery, all samples were wrapped in 2 layers of baked aluminium foil.

In the laboratory, all glassware and other materials were heated (550 °C for 3 h) and/or extensively rinsed with purified organic solvents such as acetone. Solvent cleanliness is regularly checked as part of the lab routine.

The surfaces of all samples (≥1.5 cm) were removed with an acetone-cleaned saw to eliminate potential contamination sources that are not necessarily visible with a hand-lens (e.g. hypoliths and endoliths). The cut sections were used for the preparation of petrographic thin sections. The resulting interior parts were extensively rinsed with acetone and immediately further processed for geochemical bulk analyses and fluid inclusion studies.

**Petrography and bulk analyses**. Detailed petrographic analyses were conducted on thin sections of all samples using a Zeiss SteREO Discovery.V8 stereo micro-scope linked to an AxioCam MRc5 5-megapixel camera. A subset of well-preserved samples that contained organic matter was subsequently analysed by all other methods.

Total organic carbon (TOC) contents were determined with a Leco RC612 multiphase carbon/hydrogen analyser, where carbon was measured as $CO_2$ with an infrared detector. Prior to analysis, barite was finely ground (≤50 µm) in a swing mill (MM 400, Retsch GmbH, Haan, Germany). A significant contribution of $CO_2$ from fluid inclusions to the TOC value is unlikely because most volatiles were released during grinding.

**Fluid inclusion microanalyses**. Fluid inclusions analyses were conducted on doubly polished thick sections (150–200 µm) from the interior parts of the samples.

Microthermometry measurements were performed with a Linkam THM 600 heating-freezing stage coupled to a video system (PixeLINK PL-A662, 1,3 MPixel). The stage was calibrated by using a set of synthetic fluid inclusions standards. Recorded phase transitions included (i) final melting temperatures of various solid materials ($CO_2$, $H_2S$, ice, clathrate), (ii) partial homogenisation temperatures (Th) of non-aqueous phases into aqueous or vapor phases, and (iii) total homogenisation temperatures of phases into the liquid phase [Th (total)].

Raman analysis of fluid inclusions contents was performed with a Horiba Jobin Yvon LabRam-HR 800 Raman spectrometer equipped with a 488 nm laser.

**Gas chromatography–mass spectrometry (GC–MS)**. Online (thermal decrepita-tion/desorption, TD) and offline (solid phase micro extraction, SPME) GC–MS analyses were conducted using a Varian CP-3800 GC coupled to a Varian 1200 L MS.

All mechanical tools used for sample preparation were heat-cleaned immediately before use by using a micro-torch. The cleanliness of the GC–MS system, SPME fibre, and grinding jar were assured by extensive blank analyses (see description of online and offline GC–MS analyses below). Sample runs were only performed after the immediately preceding blank chromatogram proved clean of contaminants and/or carryover products. In few instances, trace compounds were observed in pre- and post-analysis runs due to carryover (Supplementary Figs. 3, 4). These compounds

occur only in insignificant amounts and thus can be neglected. Due to previous long-term use of the GC for extract-based analyses, solvent-derived n-hexane consistently occurred in all runs, including blanks, and was eventually utilised as an internal retention time standard. White barite, being virtually devoid of organic matter, was analysed as a negative control to recognize organic volatiles from other sources, and laboratory contamination (Supplementary Fig. 5). The white barite samples were prepared and analysed in the same way as black barite samples.

For online analysis with TD-GC-MS, the Varian 1079 temperature-programmable injector of the GC–MS was equipped with a Varian ChromatoProbe adapter, which permits a direct introduction of micro vials loaded with sample material. This technique reduces the risk of contamination by rendering chemical preparation steps unnecessary. A further advantage is that only small sample amounts are required ($10^{-2}$ g range). Thin stromatolite interlayers (Fig. 1b, c) were carefully removed from the barite by using a laboratory hammer and chisel. The resulting isolated, millimetre- to centimetre-sized barite pieces were crushed with a mortar and pestle, until a grain size of roughly one millimetre was reached.

The barite grains were filled into a sample micro vial (glass; inner diameter: 1.9 mm; length: 12 mm). Using the ChromatoProbe device, the vial was directly loaded into the warm (60 °C) GC injector. To allow desorption and venting of adsorbed gases, the starting temperature was hold for 6 min at a split ratio of 50. To initiate thermal decrepitation of fluid inclusions, the GC injector was heated from 60 °C to a maximum of 250 °C at 200 °C/min and held for 3 min. Between 6 and 9.95 min, the split valve was closed to enable optimal transfer of target compounds to the GC-column. Thereafter, the split was set back to 50. Analyses of empty micro vials immediately before sample runs served as blanks to ensure cleanliness of the system.

The GC was equipped with a HP-PLOT/Q + PT fused silica capillary column coupled to a Duraguard pre-column (lengths: 30 m + 5 m, respectively; inner diameter: 0.32 mm; film thickness: 20 µm). The GC oven temperature was initially kept at 30 °C (isothermal for 9.95 min), then ramped to 250 °C (isothermal for 5 min) at 20 °C/min. Helium was used as carrier gas at a constant flow of 4 mL/min. Electron ionization mass spectra were recorded in full scan mode ($m/z$ 10–200) using a scan time of 0.5 s and electron energy of 70 eV.

Offline analysis with SPME-GC–MS involved the adsorption of volatiles contained in fluid inclusions onto a SPME fibre, followed by the subsequent injection into the GC–MS. To open the fluid inclusions, we used a custom-designed gas-tight grinding jar equipped with a septum port and a zircon milling ball (Supplementary Fig. 6). Prior to each experiment, the grinding jar was cleaned by (i) milling with combusted (550 °C, >3 h) sand for 3 × 3 min in a swing mill (MM 400, Retsch GmbH, Haan, Germany), (ii) subsequent rinsing with Millipore water, and (iii) drying at 100 °C for 2 h. Approximately 4 g of millimetre- to centimetre-sized barite fragments were prepared as described above and loaded into the cooled jar. The jar was subsequently closed, heated to 60 °C, and purged with He via the septum port for a minimum of 2 h. This ensures that any residual air and other potentially desorbed volatile compounds will be expelled from uncrushed sample surfaces. The jar was then fitted into the swing mill, and the contained fluid inclusions were opened by shaking with the zircon ball for 6 min (13 Hz) at room temperature. A pre-heated (30 min at 270 °C) and activated divinylbenzene/carboxen™/polydimethylsiloxane SPME fibre was inserted into the headspace of the jar through the septum port. Volatiles released from the fluid inclusions were allowed to adsorb onto the SPME fibre for 30 min at 50 °C (Supplementary Fig. 6).

After retraction from the jar, the syringe containing the SPME fibre was immediately placed into the GC injector to avoid airborne contamination. The GC injector temperature was kept at 250 °C, using a split ratio of 50. Upon insertion of the SPME fibre, the split valve was closed and reset to 50 after 5.1 min. The GC oven temperature was initially held at 30 °C (isothermal for 9.95 min), then ramped to 250 °C at 20 °C/min (isothermal for 10 min). MS parameters were as given above, except that the scan range was narrowed to $m/z$ 10–150.

To keep track of potential external contamination, a sequence of tests was performed prior to each sample run: (i) GC–MS blank run with no injection, (ii) GC–MS analysis of the pre-conditioned SPME fibre, (iii) SPME-GC–MS analysis of cleaned but unloaded jar, and (iv) SPME-GC–MS analysis of loaded and helium-purged jar prior to sample crushing (Supplementary Fig. 4). SPME-GC–MS analysis of crushed barite samples was only conducted if all four blanks proved clean.

Compounds were identified by comparing them to reference spectra (NIST mass spectral library) and commercially available standards (see Table 2). In case of alkylbenzenes, the elution order of isomers was additionally used for identification. Relative compound abundances were calculated separately for TD-GC–MS (150 °C, 250 °C) and SPME-GC–MS analysis. The main components ($H_2O$, $CO_2$, $H_2S$) and trace compounds were not included in these calculations. The relative abundances of most compounds vary only slightly between replicate measurements (Table 2).

**Isotope geochemistry**. Stable carbon and oxygen isotopic ratios are reported as delta values (δ) in parts per thousand (‰) relative to the Vienna Pee Dee Belemnite (VPDB) and Vienna Standard Mean Ocean Water (VSMOW), respectively.

The stable carbon isotope composition of bulk organic matter ($δ^{13}C_{TOC}$) was analysed at the Competence Center for Stable Isotope Analyses at the Georg-August-Universität Göttingen, Germany. About 11 mg of powdered and decalcified (with HCl) sample material was measured with an elemental analyser (Na-2500

CE-instruments) coupled to an isotope ratio mass spectrometer (Finnigan MAT Delta Plus). Each sample was analysed twice to monitor reproducibility. Duplicate $\delta^{13}C_{TOC}$ values are within 1.1 ‰ of each other. Acetanilide ($\delta^{13}C = -29.6$ ‰) was used for internal calibration of the instrument.

The stable carbon and oxygen isotopic composition of fluid inclusion-contained $CO_2$ ($\delta^{13}C_{CO2}$ and $\delta^{18}O_{CO2}$, respectively) was analysed using offline and online approaches. For offline analyses, samples were prepared as for SPME-GC–MS analysis. Volatiles released through grinding were retrieved from the grinding jar using a gas-tight syringe. The $CO_2$ was then cryogenically purified and analysed using a Finnigan DeltaPlus MS. For online analyses, a sample crusher was connected upstream of a gas chromatograph coupled to an NC2500 elemental analyser (Carlo Erba, Italy) and a DELTAplusXL continuous-flow isotope ratio mass spectrometer (Thermo Fisher Scientific, Germany)[71]. A volume of c. 2 cm$^3$ of barite was crushed for each analysis.

**Reporting summary**. Further information on research design is available in the Nature Research Reporting Summary linked to this article.

## Data availability
All relevant data to support the findings are provided in the paper (results section; Figs. 1–6) and the supplementary information (Supplementary Figs. 1, 3–5). GC–MS raw data are available from the corresponding author upon request.

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

## Acknowledgements
We thank Jakub Surma, Birgit Röring, Cornelia Conradt, Andreas Reimer, and Jens Dyckmans for technical and analytical support. This research was funded by the German Research Foundation (DFG grants DU 1450/4-1; DFG Priority Programme (SPP) 1833 "Building a Habitable Earth": DU 1450/3-1, DU 1450/3-2, TH 713/13-2, RE 665/42-2). This is publication number 10 of the Early Life Working Group (Department of Geobiology, University of Göttingen; Göttingen Academy of Sciences and Humanities).

## Author contributions
H.M. and V.T. conceptualized the study. H.M., J.-P.D., and J.R. performed field work and sampling. J.R. conducted petrographic analysis. A.v.d.K. carried out microthermometry and fluid inclusion microanalysis with Raman and CL. H.M. and V.T. designed and conducted the GC–MS experiments. A.P. and V.L. performed $CO_2$ stable isotope analysis. All authors contributed to discussion of results and writing.

## Funding

## Competing interests
The authors declare no competing interests.
