## [Peer Review File · Nature Communications]

REVIEWER COMMENTS

Reviewer #1 (Remarks to the Author):

Summary

This study presents petrographic, compositional, and molecular analyses of fluid inclusions in the 3.5 Ga Dresser Formation to look for evidence of primordial organic constituents needed for life in early Earth habitats. According to the manuscript, previous studies of fluid inclusions in the Dresser Formation have not included organic analyses. The objectives of the study are clearly stated and are of significance. The manuscript is clearly written, but several comments should be addressed prior to publication.

Comments

1) The manuscript presents evidence that most of the fluid inclusions are primary not secondary. However, the manuscript does not address whether the host rock rather than fluid inclusions could be the source of any of the organic molecules observed by either GC-MS methods. Both GC-MS techniques should capture volatile organic molecules of the rock matrix as well as fluid inclusions. In fact, this may explain why the SPME-GC-MS blank with uncrushed sample includes oxolane.

A negative control experiment would add confidence that the fluid inclusions are the source of the detected organic molecules. For instance, the same GC-MS experiments could be performed on a Dresser Formation sample that either did not have any fluid inclusions or only/mainly had secondary fluid inclusions. If the organic molecular composition from these experiments were distinct from the samples with primary fluid inclusions, then this would strengthen the conclusions of this study. Instead, it is not clear that the host rock matrix can be excluded as a source of volatile organic molecules. This is problematic because the host rock matrix is not a closed system like the fluid inclusions and therefore may be more prone to secondary effects.

2) The classification and composition section should briefly state the analytical methods used to get the results even if the methods details are presented at the end.

3) The manuscript should describe whether the fluid inclusion properties and inorganic compositions measured in this study agree with previous fluid inclusion studies in the Dresser Formation.

4) The TD-GC-MS results section does not describe detecting aromatic compounds, yet aromatic compounds are indicated in the chromatogram and table for TD-GC-MS analyses. The results section should be updated to reflect this.

5) In the sampling section, the manuscript describes collecting unweathered barite material from the immediate surroundings of the Dresser mine. The Pilbara has a deep weathering horizon, so it is unlikely that surface samples are completely unweathered. In the GCMS methods section, the manuscript states that the samples were prepared for TD-GC-MS from the sample interiors to avoid unweathered surface material. This statement conflicts with the previous statement regarding sampling unweathered material. The sampling section should be updated to more accurately reflect how the samples in the vicinity of the mine were selected and collected and what evidence there is for weathering or lack of weathering of the samples.

6) In the petrography and bulk analyses section, a range of TOC values is needed to better describe "organic-rich" because this can indicate a wide range of values to different readers. Similarly, the manuscript only reports a single TOC value, but more TOC values were measured according to the methods section. However, these values are not reported. They should be included somewhere in the manuscript.

7) In the fluid inclusion microanalyses methods section, the text should clarify if the thin sections were made from the exterior portions on the samples.

8) The manuscript should include a more detailed description of the blank analyses that were performed. Were solvent blanks used or what kinds of material? At what point in the sampling and sample preparation process did the blanks start? Was an empty micro vial used for a blank for TD-GC-MS analysis or was something like combusted sand loaded into it? Similarly, what kind of storage vessel was used to collect and store the samples? Were blanks taken from these storage containers?

9) The manuscript describes the GC-MS injector conditions using "split rate", but this is probably a split ratio not rate, hence the unitless value of 50. This should be fixed throughout the methods section.

10) In the isotope geochemistry methods section, the manuscript states that 11g of material was used, which seems high. It is not clear if an autosampler was used, but they typically cannot operate with that large of a sample size. Check that this is the correct unit. The methods section does not state how reproducible the isotope measurements (duplicates) were. For example, duplicate d13C TOC values were within X permil of each other. The manuscript also reports the d13C CO2 isotope data in a figure, but the d18O data or the d13C TOC data are not reported. These should be included somewhere in the manuscript.

11) Table 2 mostly uses periods as decimal places except in the last two columns where commas are used (e.g., 4,59 vs 4.59). Periods should be used throughout table 2.

12) The figure 3 legend should be simplified for clarity. The Aqueous and Non-aqueous chart labels do not provide more information than what is already included in the symbols.

13) Figure 5 description typo: add "and" before stars.

Reviewer #2 (Remarks to the Author):

Review of Mißbach et al Nature Comms, "Ingredients for microbial life preserved in 3.5 billion-year-old fluid inclusions". Simon George, 10/6/20

The paper describes the first analysis and interpretation of low molecular weight hydrocarbons and O- and S-containing compounds in fluid inclusions in a 3.5 Ga rock. The study is important, novel, and highly relevant for understanding early life metabolisms and origins. The first analysis of these types of compounds in barites opens up the possibility of further exciting research, especially related to hydrothermal systems. The paper will be widely read. I recommend publication after some changes and additions.

The key problem that requires resolution before publication relates to timing. When (at what Ts? and how long after burial?) did the barites form? If it formed a long time after deposition, after significant burial, could these fluids still hold relevance for powering the early microbial life? In the description of the rock the barites is described as "sedimentary", and it is shown that the barites is closely related to stromatolites. I think further evidence for this sedimentary nature of the barites needs to be presented, maybe with details in the supplementary section. What is the microscopic relationship of the barites to other minerals? Could the barites have formed later, diagenetically, by replacement of an earlier primary mineral? If it is sedimentary, then why are the Th values quite high, is that because of the temperature of the hydrothermal fluids? (ie warm/hot conditions at Earth's surface?). This line of argument needs to be more developed in the text, before consideration of the implications of these compounds.

The analytical work including instruments and method used appear sound, the blanks are good and properly described, and the peak identification appears to have been done appropriately.

Introduction

First paragraph: I enjoyed this summary. Suggest you enhance by mentioning the recent work on hot spring hydrothermal systems on land, e.g. work by Djokic et al 2018, and Dave Deamer et al: Damer, B. and Deamer, D. (2020) The Hot Spring Hypothesis for an Origin of Life. *Astrobiology* 20, 429-452.

Deamer, D., Damer, B. and Kompanichenko, V. (2019) Hydrothermal Chemistry and the Origin of Cellular Life. *Astrobiology* 19, 1523-1537.

Results.

TOC is given as one number, no range. At L257, you write "contents were determined", implying multiple analyses which is what I would expect. What is range or \pm ?

Discussion.

L151. It would be more helpful for the reader if some example ranges were given, especially for temperature.

Methods

"n-hexane steadily occurred in all blank runs and was eventually utilised as an internal standard." Instead of "steadily, suggest "consistently". Where was the hexane stored in the instrument?, normally it is so volatile that it disappears rapidly once surfaces and components are heated to normal operating Ts? Assuming variable flux of n-hexane into your sample runs (which I would imagine would be the case if it was coming from somewhere that you would have liked it not to come from), I suggest you could not wisely use it as an internal standard for quantification, but rather maybe you used it effectively as a retention time standard? (especially as you only provide relative abundances, which seems the right thing to do.

Minor edits and suggestions

L36 "A widely received hypothesis" Strange phrase, suggest change to something like "One hypothesis"

L45/17 Age of Dresser Formation: 3.5 or 3.49 Ga? Suggest be consistent, both have "c."

L63 "as main inorganic constituents". Add "the"

L66 "biologically-relevant"

L75 "ranging from white over grey to black." to "ranging from white and grey to black."

L92 "non-aqueous fluid inclusions, have no detectable water." Remove ,

L96 "usually show" to "usually contain"

L97 spell out the three abbreviations

L135 "toluenes" is slang, please change to alkylbenzenes

L139 "to the SPME" to "onto the SPME"

L175 advanced heating is a strange phrase, maybe change to heating to higher temperatures

L176 thermally-driven

L178 much higher to much greater

L182 "inclusions derived" to "inclusions may have been derived". Suggest soften this a bit, as you promote the possible biological origin in next para.

L188 "as carbon" to "as a carbon"

L237 "picture on the" to "picture of the":

Table 2: last 2 columns, change , to . in numbers

1,4-Xylene (p), etc. Incorrect usage. It is either p-xylene or 1,4-dimethylbenzene. Also in Fig. S2

L254 analysis was to analyses were

L258 an infrared

L273 use same hyphen type in full name and abbreviation

L277 change "employing a micro-torch." to "by using a micro-torch."

L277 jar were

L295 "max. 250°C with 200°C/min and held constant for 3 min." to "a maximum of 250°C at 200°C/min and held constant for 3 min."

L306 "sea sand" beach sand?

L318 up of airborne

L328 "methylbenzenes". Based on Table 2, I think you mean alkylbenzenes (ie you used RTs to determine 1,2,3-TMB vs other TMBs).

L564 and 571 "labels as" to "labels are"

Simon George, 10/6/20

Response to the reviewers' comments

Please note that the stated line numbers of the original reviewer comments refer to the old manuscript, whereas the line numbers given in the reply (in brackets) refer to the revised manuscript with tracked changes. Original comments are given in bold.

Reviewer #1:

Summary

This study presents petrographic, compositional, and molecular analyses of fluid inclusions in the 3.5 Ga Dresser Formation to look for evidence of primordial organic constituents needed for life in early Earth habitats. According to the manuscript, previous studies of fluid inclusions in the Dresser Formation have not included organic analyses. The objectives of the study are clearly stated and are of significance. The manuscript is clearly written, but several comments should be addressed prior to publication.

Comments

1) The manuscript presents evidence that most of the fluid inclusions are primary not secondary. However, the manuscript does not address whether the host rock rather than fluid inclusions could be the source of any of the organic molecules observed by either GC-MS methods. Both GC-MS techniques should capture volatile organic molecules of the rock matrix as well as fluid inclusions. In fact, this may explain why the SPME-GC-MS blank with uncrushed sample includes oxolane. A negative control experiment would add confidence that the fluid inclusions are the source of the detected organic molecules. For instance, the same GC-MS experiments could be performed on a Dresser Formation sample that either did not have any fluid inclusions or only/mainly had secondary fluid inclusions. If the organic molecular composition from these experiments were distinct from the samples with primary fluid inclusions, then this would strengthen the conclusions of this study. Instead, it is not clear that the host rock matrix can be excluded as a source of volatile organic molecules. This is problematic because the host rock matrix is not a closed system like the fluid inclusions and therefore may be more prone to secondary effects.

→ We agree that the host rock matrix is not a closed but an open system. However, many of the low molecular weight compounds we found to be present are highly volatile (especially gases). In open systems, such volatiles readily escape and are thus extremely unlikely to be preserved. In closed systems, in contrast, volatiles cannot escape directly and thus might be preserved. The presence of low molecular weight volatile compounds in the analysed samples thus requires the presence of closed systems – such as the petrographically observed fluid inclusions.

The fact that the detected compounds derive from fluid inclusions and thus from closed systems is further strengthened by the following arguments:

- Consistency of data obtained by independent analytical techniques (Raman spectroscopy vs. GC-MS). The mean abundance of compounds in single fluid inclusions as revealed by Raman spectroscopy are well in line with relative abundances obtained by GC-MS (compare Table 1 and 2, see further Figs. 4 and 5).
- Consistency of data with previous studies. The observed inorganic fluid inclusion contents are consistent with data presented in earlier studies (Harris et al., 2009; Rankin and Shepherd, 1978; compare with Table 2 and Fig. 4).

- Temperature dependency of product yields. Higher temperature analyses result in higher product yields (TD-GC-MS 150 °C vs. TD-GC-MS 250°C vs. SPME-GC-MS). This is not to be expected in case of a rock matrix source since volatiles can readily escape at low temperatures from such open systems. Fluid inclusions (i.e., closed systems), in contrast, are much more stable and decrepitate only at higher temperatures, which plausibly explains the observed temperature dependency of product yields (lines 142 ff., Table 2, Figs. 4 and 5).
- The absence of molecular contamination indications. Bulk pyrolysis-GC-MS analyses at 340 °C for 10 s on a DB-5 column almost exclusively yielded CO₂ and SO₂, while higher molecular weight molecules such as *n*-alkanes were not detected. The oxolane detected in the SPME-GC-MS blank was also found in some empty GC-MS runs and therefore assuredly due to carryover.

Together, these multiple lines of evidence strongly suggest that the analysed compounds derived from the fluid inclusions (closed systems) and not from the rock matrix (open system). However, to further corroborate this, we followed the reviewer's suggestion and additionally analysed a white barite (virtually devoid of organic matter) as a negative control. We found that the white barite indeed only contained traces of CO₂ and H₂O (new result presented in lines 329 to 331. in the methods section and in the supplement, Fig. S.5). We also analysed black barite samples in parallel as positive control. The results of these analyses are identical to our earlier findings (presented in Table 2 and Figs. 4, 5). In combination, the negative and positive control experiments thus add confidence that the detected organic molecules derived from the fluid inclusions and not the rock matrix.

2) The classification and composition section should briefly state the analytical methods used to get the results even if the methods details are presented at the end.

- ➔ We now shortly state the methods used to analyse the fluid inclusions at the beginning of the respective subsection (line 96).

3) The manuscript should describe whether the fluid inclusion properties and inorganic compositions measured in this study agree with previous fluid inclusion studies in the Dresser Formation.

- ➔ The fluid inclusion properties and inorganic compositions obtained in our study largely agree with those of earlier studies. The manuscript was extended accordingly (lines 181 to 186)

4) The TD-GC-MS results section does not describe detecting aromatic compounds, yet aromatic compounds are indicated in the chromatogram and table for TD-GC-MS analyses. The results section should be updated to reflect this.

- ➔ We added benzene to the TD-GC-MS results (lines 141 and 142).

5) In the sampling section, the manuscript describes collecting unweathered barite material from the immediate surroundings of the Dresser mine. The Pilbara has a deep weathering horizon, so it

is unlikely that surface samples are completely unweathered. In the GCMS methods section, the manuscript states that the samples were prepared for TD-GC-MS from the sample interiors to avoid unweathered surface material. This statement conflicts with the previous statement regarding sampling unweathered material. The sampling section should be updated to more accurately reflect how the samples in the vicinity of the mine were selected and collected and what evidence there is for weathering or lack of weathering of the samples.

→ We see that these parts are confusingly written and revised the text accordingly. In the sampling section, we provide additional information on how samples were collected (lines 281 ff.). Furthermore, we replaced “Unweathered...” with “Fresh...” to avoid confusion (line 279). The information provided in the GC method section (“..., avoiding weathered surfaces”) refers to oxidised and brittle stromatolite layers that are associated with the barites. We rephrased the passage to make this clearer (“Thin stromatolite interlayers were carefully removed from the barite by using a laboratory hammer and chisel”; lines 336 to 338).

6) In the petrography and bulk analyses section, a range of TOC values is needed to better describe “organic-rich” because this can indicate a wide range of values to different readers. Similarly, the manuscript only reports a single TOC value, but more TOC values were measured according to the methods section. However, these values are not reported. They should be included somewhere in the manuscript.

→ In this context, “organic-rich” was meant to describe samples that regularly showed organic content during petrographic analysis. Only such samples were analysed by other, more time-consuming methods (e.g. GC-MS). However, we understand the reviewer’s concern that “organic-rich” is not precise enough and therefore changed the wording to make this point clearer to the reader (line 298). The TOC values from 5 different analyses were very close together (0.309 – 0.314 wt%). To express this we would have to report 3 decimal places which would, however, suggest an unrealistic instrumental precision. Therefore, we decided to display the mean TOC value (0.31 wt%) and give additional statistical information (N= 5; standard deviation = 0.002) (lines 155 and 156).

7) In the fluid inclusion microanalyses methods section, the text should clarify if the thin sections were made from the exterior portions on the samples.

→ The thick sections were made from the interior part of the samples. We now state this in the manuscript (lines 306 and 307). Thin sections from the outer parts were only used for petrographic analysis.

8) The manuscript should include a more detailed description of the blank analyses that were performed. Were solvent blanks used or what kinds of material? At what point in the sampling and sample preparation process did the blanks start? Was an empty micro vial used for a blank for TD-GC-MS analysis or was something like combusted sand loaded into it? Similarly, what kind of storage vessel was used to collect and store the samples? Were blanks taken from these storage containers?

- In our study, solvents were only used for cleaning the outer surfaces of the original rock sample pieces, and some tools during processing. These solvents are regularly checked for cleanliness as part of the lab routine. Therefore, we did not perform separate solvent blanks. Aluminium foil is extensively used in our lab to protect samples from contamination during sampling, processing and storage. During production, the aluminium foil is exposed to temperatures >650 °C and therefore considered clean. The blanks that are described in the GC-MS method section were done as contamination control during the analysis. The white barite analysed (see comment #1) serves also as a negative control for contamination during sampling and processing (see Fig. S.5 in the supplement and lines 329 to 331). The following information about storage containers and blank analyses were added to the manuscript.
- Samples were wrapped in aluminium foil (lines 284 and 285)
 - Solvents are regularly checked for cleanliness as part of the lab routine (lines 287 and 288)
 - Empty vial analyses were performed as blanks for the online technique (TD-GC-MS; lines 348 and 349)
 - A detailed description of the blank analysis procedure for the offline technique (SPME-GC-MS; lines 379 to 384)

9) The manuscript describes the GC-MS injector conditions using “split rate”, but this is probably a split ratio not rate, hence the unitless value of 50. This should be fixed throughout the methods section.

- Changed as recommended (line 374).

10) In the isotope geochemistry methods section, the manuscript states that 11g of material was used, which seems high. It is not clear if an autosampler was used, but they typically cannot operate with that large of a sample size. Check that this is the correct unit. The methods section does not state how reproducible the isotope measurements (duplicates) were. For example, duplicate d13C TOC values were within X permil of each other. The manuscript also reports the d13C CO2 isotope data in a figure, but the d18O data or the d13C TOC data are not reported. These should be included somewhere in the manuscript.

- Indeed, 11 mg and not g were used. We changed the manuscript accordingly (line 397). The errors for the stable isotope analyses as well as the $\delta^{18}\text{O}$ data are given in the results section (lines 156 to 159). Additionally, we added information about the reproducibility of the $\delta^{13}\text{C}$ TOC analysis to the methods section (lines 399 and 400). Furthermore, we recognised a typo in the given mean $\delta^{13}\text{C}$ TOC value in the results section which was corrected (27.6 +/- 0.6 instead of 28.2 +/- 0.5; line 156). This change has no influence on the interpretation.

11) Table 2 mostly uses periods as decimal places except in the last two columns where commas are used (e.g., 4,59 vs 4.59). Periods should be used throughout table 2.

- Changed accordingly.

12) The figure 3 legend should be simplified for clarity. The Aqueous and Non-aqueous chart labels do not provide more information than what is already included in the symbols.

→ We have simplified the Fig. 3 legend and removed the chart labels.

13) Figure 5 description typo: add “and” before stars.

→ Changed accordingly, also in the caption to Fig. 4 (lines 650 and 659).

Reviewer #2:

Review of Mißbach et al Nature Comms, “Ingredients for microbial life preserved in 3.5 billion-year-old fluid inclusions”. Simon George, 10/6/20

The paper describes the first analysis and interpretation of low molecular weight hydrocarbons and O- and S-containing compounds in fluid inclusions in a 3.5 Ga rock. The study is important, novel, and highly relevant for understanding early life metabolisms and origins. The first analysis of these types of compounds in barites opens up the possibility of further exciting research, especially related to hydrothermal systems. The paper will be widely read. I recommend publication after some changes and additions.

The key problem that requires resolution before publication relates to timing. When (at what Ts and how long after burial?) did the barites form? If it formed a long time after deposition, after significant burial, could these fluids still hold relevance for powering the early microbial life? In the description of the rock the barites is described as “sedimentary”, and it is shown that the barites is closely related to stromatolites. I think further evidence for this sedimentary nature of the barites needs to be presented, maybe with details in the supplementary section. What is the microscopic relationship of the barites to other minerals? Could the barites have formed later, diagenetically, by replacement of an earlier primary mineral? If it is sedimentary, then why are the Th values quite high, is that because of the temperature of the hydrothermal fluids? (ie warm/hot conditions at Earth’s surface?). This line of argument needs to be more developed in the text, before consideration of the implications of these compounds. The analytical work including instruments and method used appear sound, the blanks are good and properly described, and the peak identification appears to have been done appropriately.

→ Barites of the Dresser Formation occur either in form of primary bedded layers or in form of hydrothermal veins. All samples analysed herein are from bedded layers. A primary origin of these barites is supported by field and petrographic evidence. For instance, the originally sulfidic stromatolite interbeds are still largely intact and show no indications for layer disruption or a progressive replacement by barite (Fig. 1b, c). Furthermore, detailed thin section petrography revealed no red-flags for a possible secondary replacement of pre-existing minerals (e.g., pseudomorphs, crystal intergrowth or fragmentation) (Fig. 2). These findings are in excellent accordance with earlier studies that interpreted bedded barites of the Dresser Formation as primary hydrothermal sediments (e.g. Nijman et al., 1998; Van Kranendonk and Pirajno, 2004; Van Kranendonk, 2006; Harris et al. 2009; Djokic et al., 2017).

The Th values from fluid inclusions in the black barite certainly represent the temperature of the hydrothermal fluids at the time of mineral precipitation. However, this does not mean that the environments were uninhabitable. Taking modern hydrothermal environments as example, it must be assumed the hot fluids rapidly cooled as they mixed with cooler surface waters, resulting in habitable conditions (particularly for thermophilic organisms). It therefore appears plausible that microbial life thrived under such conditions. Notably, some of the authors referenced in the previous paragraph emphasize the close relation between hydrothermal fluids and life in the Dresser Formation.

We describe and discuss features that are relevant to the primary nature of the analysed barites in the revised results and discussion sections (lines 84 to 86 and 164 to 170, respectively).

Introduction

First paragraph: I enjoyed this summary. Suggest you enhance by mentioning the recent work on hot spring hydrothermal systems on land, e.g. work by Djokic et al 2018, and Dave Deamer et al: Damer, B. and Deamer, D. (2020) The Hot Spring Hypothesis for an Origin of Life. Astrobiology 20, 429-452.

Deamer, D., Damer, B. and Kompanichenko, V. (2019) Hydrothermal Chemistry and the Origin of Cellular Life. Astrobiology 19, 1523-1537.

→ We added the suggested references to the manuscript (line 38 and lines 436 ff.).

Results.

TOC is given as one number, no range. At L257, you write “contents were determined”, implying multiple analyses which is what I would expect. What is range or ±?

→ The TOC values from 5 different analyses were very close together (0.309 – 0.314 wt%). To express this we would have to report 3 decimal places which would, however, suggest an unrealistic instrumental precision. Therefore, we decided to display the mean TOC value (0.31 wt%) and give additional statistical information (N= 5; standard deviation = 0.002) (lines 155 and 156; see also comment #6 by Rev. 1).

Discussion.

L151. It would be more helpful for the reader if some example ranges were given, especially for temperature.

→ The temperature range for Th (total) was added (lines 175 and 176).

Methods

“n-hexane steadily occurred in all blank runs and was eventually utilised as an internal standard.” Instead of “steadily, suggest “consistently”. Where was the hexane stored in the instrument?, normally it is so volatile that it disappears rapidly once surfaces and components are heated to normal operating Ts? Assuming variable flux of n-hexane into your sample runs (which I would imagine would be the case if it was coming from somewhere that you would have liked it not to come from), I suggest you could not wisely use it as an internal standard for quantification, but rather maybe you used it effectively as a retention time standard? (especially as you only provide relative abundances, which seems the right thing to do.

- Obviously, the *n*-hexane accumulated during previous long-term use of the instrument for extract-based analyses (using *n*-hexane as the solvent for sample injection). It has been residing in the non-heated parts of the instrument that are connected to the injector, probably in the stainless steel split lines, from where small amounts are being released upon any valve switching sequence (i.e. at the starting point of any run). As the reviewer indicated, the steadily appearing *n*-hexane peak cannot be used for quantification, but it turned out to be very useful for our study as a retention time standard. We clarified this and changed the wording according to the reviewer's suggestions (line 329; Figs. 4, 5, S.3, S.4, S.5; Table 2).

Minor edits and suggestions

L36 "A widely received hypothesis" Strange phrase, suggest change to something like "One hypothesis"

- We changed the wording as suggested (line 41).

L45/17 Age of Dresser Formation: 3.5 or 3.49 Ga? Suggest be consistent, both have "c."

- The age was consistently changed to c. 3.5 Ga (lines 50, 73)

L63 "as main inorganic constituents". Add "the"

- Done (line 69).

L66 "biologically-relevant"

- Changed as suggested (line 72).

L75 "ranging from white over grey to black." to "ranging from white and grey to black."

- Changed accordingly (line 81).

L92 "non-aqueous fluid inclusions, have no detectable water." Remove ,

- We removed this part (line 102).

L96 "usually show" to "usually contain"

- Changed accordingly (line 106).

L97 spell out the three abbreviations

- Done (line 107).

L135 "toluenes" is slang, please change to alkylbenzenes

- Changed accordingly (line 148).

L139 “to the SPME” to “onto the SPME”

→ Changed accordingly (line 153).

L175 advanced heating is a strange phrase, maybe change to heating to higher temperatures

→ Changed accordingly (line 205).

L176 thermally-driven

→ Changed accordingly (line 206).

L178 much higher to much greater

→ Changed accordingly (line 208).

L182 “inclusions derived” to “inclusions may have been derived”. Suggest soften this a bit, as you promote the possible biological origin in next para.

→ We changed the wording as suggested (line 212).

L188 “as carbon” to “as a carbon”

→ Changed accordingly (line 219).

L237 “picture on the” to “picture of the”:

→ Changed accordingly (line 269).

Table 2: last 2 columns, change , to . in numbers

→ Changed accordingly.

1,4-Xylene (p), etc. Incorrect usage. It is either p-xylene or 1,4-dimethylbenzene. Also in Fig. S2

→ We changed the molecules names in table 2 and Fig. S2 as recommended.

L254 analysis was to analyses were

→ Changed accordingly (line 296).

L258 an infrared

→ Changed accordingly (line 301).

L273 use same hyphen type in full name and abbreviation

→ Changed accordingly (line 317).

L277 change “employing a micro-torch.” to “by using a micro-torch.”

→ Changed accordingly (line 277).

L277 jar were

→ Changed accordingly (line 322).

L295 “max. 250°C with 200°C/min and held constant for 3 min.” to “a maximum of 250°C at 200°C/min and held constant for 3 min.”

→ The wording was changed accordingly (lines 345 and 346).

L306 “sea sand” beach sand?

→ We changed the wording from “sea sand” to “sand” (line 360).

L318 up of airborne

→ We changed the wording (line 373).

L328 “methylbenzenes”. Based on Table 2, I think you mean alkylbenzenes (ie you used RTs to determine 1,2,3-TMB vs other TMBs).

→ “Methylbenzenes” was changed to “alkylbenzenes” (line 386).

L564 and 571 “labels as” to “labels are”

→ Changed accordingly (see captions to Figs. 4 and 5; lines 649 and 657, respectively).

REVIEWER COMMENTS

Reviewer #1 (Remarks to the Author):

Review of revised Mißbach et al, “Ingredients for microbial life preserved in 3.5 billion-year-old fluid inclusions”

The revised version adequately addresses most of my previous comments, and therefore it is suitable for publication.

While the manuscript presents a reasonable case that the fluid inclusions are the likely source for the volatile organic compounds, volatile organic components can reside in the rock matrix in pores and in association with surfaces. The additional analysis of the white barite helps to mitigate the issue of whether the rock matrix is a source, but it does not completely rule out the rock matrix as a possible additional source of volatile organic compounds in the GC-MS analyses. A more balanced approach may be to expand the section in lines 188-192, presenting all of the arguments for why the fluid inclusions are the most likely source of the volatile organics while some degree of contribution from the rock matrix to the GC-MS analyses cannot be entirely precluded.

In lines 315-317, the manuscript should clarify whether the white barite sample was treated in the same way as the “organic-rich” barites, including sawing, powdering, loading in sample micro vials for TD-GC-MS, and processing for SPME-GC-MS analyses. This additional clarification would help to more fully flesh out the contamination control of the study.

Reviewer #2 (Remarks to the Author):

Review of revised Mißbach et al Nature Comms, "Ingredients for microbial life preserved in 3.5 billion-year-old fluid inclusions". Simon George, 26/10/20

I am happy with the new sections added about the primary nature of the barites, including the new paragraph "data integrity". These substantially improve the paper.

L286. Change word "virtually" in this new sentence

A subset of well preserved samples that virtually contained organic matter was subsequently analysed by all other methods.

Regarding author comment "During production, the aluminium foil is exposed to temperatures >650 °C and therefore considered clean". That is true about production, but in my lab we have found that different rolls of Al foil as received can sometime be variably contaminated, probably from lubricants used for rolling the foil. Authors beware!! We regularly carry out Al foil blanks to check on this issue.

I am happy that all the other minor changes were done. I recommend acceptance of this paper.

Simon George, 26/10/20

Response to the reviewers' comments – 2nd review

Please note that the stated line numbers of the original reviewer comments refer to the old manuscript, whereas the line numbers given in the reply (in brackets) refer to the revised manuscript with tracked changes. Original comments are given in bold.

Reviewer 1:

Review of revised Mißbach et al, “Ingredients for microbial life preserved in 3.5 billionyear-old fluid inclusions”

The revised version adequately addresses most of my previous comments, and therefore it is suitable for publication.

While the manuscript presents a reasonable case that the fluid inclusions are the likely source for the volatile organic compounds, volatile organic components can reside in the rock matrix in pores and in association with surfaces. The additional analysis of the white barite helps to mitigate the issue of whether the rock matrix is a source, but it does not completely rule out the rock matrix as a possible additional source of volatile organic compounds in the GC-MS analyses. A more balanced approach may be to expand the section in lines 188-192, presenting all of the arguments for why the fluid inclusions are the most likely source of the volatile organics while some degree of contribution from the rock matrix to the GC-MS analyses cannot be entirely precluded.

We expanded the respective section as recommended (lines 190 to 197), presenting all the arguments provided in our previous response letter.

In lines 315-317, the manuscript should clarify whether the white barite sample was treated in the same way as the “organic-rich” barites, including sawing, powdering, loading in sample micro vials for TD-GC-MS, and processing for SPME-GC-MS analyses. This additional clarification would help to more fully flesh out the contamination control of the study.

We clarified this in the respective section by adding the following sentence (line 322 f.): “The white barite samples were prepared and analyzed in the same way as black barite samples.”

Reviewer 2:

Review of revised Mißbach et al Nature Comms, “Ingredients for microbial life preserved in 3.5 billion-year-old fluid inclusions”. Simon George, 26/10/20

I am happy with the new sections added about the primary nature of the barites, including the new paragraph “data integrity”. These substantially improve the paper.

L286. Change word “virtually” in this new sentence

A subset of well preserved samples that virtually contained organic matter was subsequently analysed by all other methods.

We deleted the word “virtually” (line 291) since it is clear from context (including the section heading) that the organic matter was observed by petrographic analysis.

Regarding author comment “During production, the aluminium foil is exposed to temperatures >650 °C and therefore considered clean”. That is true about production, but in my lab we have found that different rolls of Al foil as received can sometime be variably contaminated, probably from lubricants used for rolling the foil. Authors beware!! We regularly carry out Al foil blanks to check on this issue.

Thank you very much for the warning, we very much appreciate that. In fact, we regularly performed Al foil blanks as part of our lab routine, and these have been constantly negative for organic contamination.

I am happy that all the other minor changes were done. I recommend acceptance of this paper.